# Quantitative Phase Imaging of Spreading Fibroblasts Identifies the Role of Focal Adhesion Kinase in the Stabilization of the Cell Rear

**DOI:** 10.3390/biom10081089

**Published:** 2020-07-22

**Authors:** Olga Ramaniuk, Zuzana Klímová, Tomáš Groušl, Tomáš Vomastek

**Affiliations:** Laboratory of Cell Signalling, Institute of Microbiology of the Czech Academy of Sciences, 142 00 Prague, Czech Republic; ramaniuk@biomed.cas.cz (O.R.); zuzana.klimova@biomed.cas.cz (Z.K.)

**Keywords:** coherence-controlled holographic microscopy, quantitative phase imaging, cell dry mass, focal adhesion kinase, Rack1, extracellular matrix, cell adhesion, cell spreading, front–rear polarity

## Abstract

Cells attaching to the extracellular matrix spontaneously acquire front–rear polarity. This self-organization process comprises spatial activation of polarity signaling networks and the establishment of a protruding cell front and a non-protruding cell rear. Cell polarization also involves the reorganization of cell mass, notably the nucleus that is positioned at the cell rear. It remains unclear, however, how these processes are regulated. Here, using coherence-controlled holographic microscopy (CCHM) for non-invasive live-cell quantitative phase imaging (QPI), we examined the role of the focal adhesion kinase (FAK) and its interacting partner Rack1 in dry mass distribution in spreading Rat2 fibroblasts. We found that FAK-depleted cells adopt an elongated, bipolar phenotype with a high central body mass that gradually decreases toward the ends of the elongated processes. Further characterization of spreading cells showed that FAK-depleted cells are incapable of forming a stable rear; rather, they form two distally positioned protruding regions. Continuous protrusions at opposite sides results in an elongated cell shape. In contrast, Rack1-depleted cells are round and large with the cell mass sharply dropping from the nuclear area towards the basal side. We propose that FAK and Rack1 act differently yet coordinately to establish front–rear polarity in spreading cells.

## 1. Introduction

The directional migration of mesenchymal cells is an adhesion-dependent process characterized by a repetition of the cell’s front protrusion, rear retraction, and cell body forward translocation [1]. Efficient migration requires the establishment and maintenance of front–rear polarity that allows a migrating cell to differently regulate the processes at leading and trailing edges. Persistently migrating cells display a polarized shape with a broad protruding cell front and a non-protruding cell rear. The establishment and perseverance of these functionally and morphologically distinct regions relies on spatial differences in Rho GTPases activities, actin cytoskeleton organization, and adhesion size and dynamics [1,2]. 

The protrusions at the cell front are the consequence of actin polymerization that generates the forces pushing the cell membrane forward. Rho-family GTPase Rac is the central activator of actin polymerization; however, Cdc42 and RhoA are also activated in this region and participate in the regulation of actin polymerization and cell edge dynamics [3,4,5]. Active Rac and Rho GTPases in turn activate the Arp2/3 complex and other actin nucleators including formins that increase actin filament meshwork formation and membrane protrusions [6,7]. Membrane protrusions are coupled to the extracellular matrix (ECM) through small dot-like integrin mediated adhesions [2,7,8]. These stationary yet dynamic adhesions stabilize the protrusions and are essential to transform part of the polymerization mechanical force into a protrusion [9]. The protruding leading edge and adhesions also serve as the sites of the formation of dorsal stress fibers, transverse actin arcs, and perinuclear actin fibers [10,11,12]. 

The cell sides and rear are, in contrast to the leading edge, non-protruding, and contain contractile actomyosin stress fibers with their ends anchored in large, elongated adhesions [2,8]. Stress fibers are present at the ventral side of the cell and they are preferentially aligned with the polarity axis [13,14,15]. A specific subset of ventral stress fibers underlying the cell sides at the periphery of the cell are under high tension [16,17]. The ends of the peripheral stress fibers terminate in stable elongated adhesions at the cell rear [13,18]. These long-lived adhesions stabilize the cell tail and promote persistent cell migration [19]. During migration, the destabilization of rear adhesions is a prerequisite for rear retraction and forward cell translocation [20,21]. Rear retraction and its inward movement are accompanied by stress fiber shortening and adhesion sliding, followed by their disassembly [22,23,24,25]. Actomyosin contractility and cell rear retraction involve RhoA GTPase, which is active at this location [1,2,26].

Asymmetric organization of the actin cytoskeleton and adhesions is relayed to the cell interior to further propagate the polarized state of the cell. Intracellular polarization is associated with a significant redistribution of cell mass, namely the nucleus. In polarizing cells, the nucleus moves to the rear of the cell [27,28,29,30,31,32]. In some cell types, additional subtypes of stress fibers, dorsal fibers, transvers arcs, and perinuclear actin fibers induce rotational movement of the nucleus and align it with the axis of migration [12,29]. Movement of the nucleus to the cell rear allows the microtubule-organizing center (MTOC) and Golgi apparatus to move to the cell center between the nucleus and the leading edge [27]. When the nucleus is at the cell’s rear it acts as a steric hindrance for microtubules and ensures that microtubules are preferentially oriented towards the cell front. The polarized microtubule network then differentially affects processes at the leading and trailing edges through polarized vesicular trafficking and cargo delivery, regulation of adhesion dynamics, and the regulation of Rho GTPases [33,34,35]. 

The establishment of migratory polarity is commonly imposed by the gradient of external directional signals, such as chemotactic or mechanical stimuli, which induce actin polymerization at the cell front [1]. Migratory cell polarity can also develop spontaneously in homogenous environments in the absence of directional signals, e.g., in spreading fibroblasts or in stationary fish keratocytes that initiate motility. Spontaneous migratory polarization in these cells is predominantly initiated by establishing the cell rear that precedes the specification of the cell front [36,37,38,39,40]. In chick fibroblasts and keratocytes, the formation of the cell rear is initiated either by a break in the peripheral actin meshwork [36] or by a local increase in actomyosin tension [37], respectively. This is followed by retraction of the cell edge. In mammalian cells, actomyosin contractility and the assembly of thick peripheral actomyosin bundles at the cell rear correlates with the appearance of elongated and stable focal adhesions and a reduction in protrusions at the cell rear [38,39,40]. On the other hand, the symmetrical distribution of adhesive contacts can keep cells in the non-polarized state [40,41]. Local inhibition of adhesive contacts on one side of non-polarized cells then leads to the formation of the cell rear at this region and induces migration in the opposite direction [41,42]. Since the reorganization of the actin cytoskeleton is linked to focal adhesions, focal adhesions thus may be a key factor in cell spontaneous polarization. The characterization of focal adhesion components in the context of spontaneous cell polarization is hence a prerequisite for the understanding of how cell asymmetry is developed. 

Focal adhesions function as signaling hubs that affect cytoskeletal organization and, consequently, cell shape and polarity. The non-receptor tyrosine kinase FAK (focal adhesion kinase) is a central mediator of integrin signaling. FAK is activated and recruited to adhesion plaques in an integrin-dependent manner. FAK associates with the kinase Src, forming the FAK/Src heterodimer which phosphorylates a plethora of adhesion proteins, altering the function of focal adhesions. In addition to the kinase role, FAK functions as a scaffold protein and by recruiting a diverse set of proteins, it regulates multiple downstream signaling pathways [43,44,45]. The important interacting partners of FAK are proteins of the RhoGAPs (GTPase activating proteins) and RhoGEFs (guanine nucleotide exchange factors) families. The recruitment of RhoGAPs and RhoGEFs modulates the activity of Rho GTPases and affects focal adhesions and actin dynamics [45]. FAK also activates the extracellular signal-regulated kinase (ERK) pathway that increases focal adhesion disassembly [46,47]. FAK protein level and activity is highest at leading edge focal adhesions and gradually decreases towards the rear [29,48]. Its activity thus correlates with the rate of focal adhesion disassembly. Concordantly with its role in the regulation of Rho GTPases, cytoskeleton, and adhesion dynamics, FAK plays an important role in the establishment and maintenance of intracellular polarity in cells polarizing to the wound [29,49,50,51,52]. However, the functions of FAK during spontaneous polarization in spreading cells remains unknown.

One of the FAK client proteins is the scaffold protein Rack1 (receptor for activated C kinase 1), which transmits integrin and FAK signaling towards downstream responses including cell migration [47,53,54,55,56]. Previously, we have shown that Rack1 together with FAK are required for efficient ERK activation by integrin engagement to the ECM and active ERK localization to focal adhesions [40,47]. In spreading Rat2 fibroblasts, Rack1 and ERK are required for spontaneous cell polarization as Rack1-depleted cells are unable to establish a non-protruding region, and steadily spread with radial symmetry. Continuous omnidirectional spreading ultimately results in a round, “fried-egg”-like non-polarized phenotype [40]. 

Here we use quantitative phase imaging as an unbiased approach to examine the role of FAK and Rack1 in the spontaneous establishment of front–rear polarity in spreading fibroblasts. Quantitative phase images are reconstructed from a hologram obtained by a coherence-controlled holographic microscope. This non-invasive label-free microscopy technique enables monitoring of phase distribution, i.e., the phase shift of a light wave upon passing through a cell. Since the phase shift has been shown to be invariant to cellular water content [57], the acquired signal is directly proportional to cell dry mass. Cell dry mass, i.e., the non-aqueous content of a cell, consisting namely of proteins, nucleic acids, lipids, sugars, and so forth, delineates the major structures of a cell (e.g., cell boundaries, intracellular compartments). Besides the direct quantification of cell dry mass distribution and its changes in time, coherence-controlled holographic microscopy generates artefact-free images of superior contrast quality facilitating subsequent image analyses [58].

Characterization of spreading cells showed that FAK-depleted cells develop bipolar symmetry, but are incapable of defining the cell front and rear. Rather, they form two distally positioned protruding regions, resulting in elongated, bipolar cells. The phenotypes of FAK- and Rack1-depleted cells are different as Rack1-depleted cells spread with radial symmetry without establishing the bipolar phenotype. These results suggest that FAK and Rack1 differently regulate the development of the cell rear and thus the establishment of front–rear polarity in spreading cells.

## 2. Materials and Methods

### 2.1. Antibodies and Reagents

The following antibodies were used for immunofluorescence: Primary antibodies *anti*-FAK, rabbit polyclonal 1:150 (Cat.#06-543, Upstate Biotechnology, Lake Placid, NY, USA) and *anti*-paxillin, clone 5H11, mouse monoclonal 1:150 (Cat.#05-417, Upstate Biotechnology, Lake Placid, NY, USA); secondary antibodies Alexa Fluor 488, goat *anti*-mouse 1:200 (Cat.#A11029, ThermoFisher Scientific, Waltham, MA, USA) and Alexa Fluor 488, goat *anti*-rabbit 1:200 (Cat.#A11034, ThermoFisher Scientific, Waltham, MA, USA). Rhodamine Phalloidin R415 (ThermoFisher Scientific, Waltham, MA, USA) 1:300 was used for visualization of actin.

The western blot technique was performed with the following antibodies: *anti*-FAK, clone 4.47, mouse monoclonal 1:1000 (Cat.#05-537, MERCK-Millipore, Darmstadt, Germany); *anti*-Rack1 mouse monoclonal, clone B3, 1:1000 (Cat.#sc-17754, Santa-Cruz Biotechnology, Dallas, TX, USA); *anti*-RSK (ribosomal protein S6 kinase) mouse monoclonal, 1:2000 (Cat.#610225, BD Transduction Laboratories, San Francisco, CA, USA); secondary antibodies HRP-conjugated goat *anti*-mouse IgG 1:80.000 (A9044, Sigma-Aldrich/Merck, Darmstadt, Germany) and goat *anti*-rabbit 1:5.000 (A8275, Sigma-Aldrich/Merck, Darmstadt, Germany).

### 2.2. Cell Culture, siRNA, and Plasmid Transfection

Rat2 cells were cultivated in Dulbecco’s modified Eagle’s medium (DMEM) (Sigma-Aldrich/Merck, Darmstadt, Germany) supplemented with 10% fetal bovine serum (ThermoFisher Scientific, Waltham, MA, USA) in a humidified atmosphere with 5% CO_2_ at 37 °C.

siRNAs were transiently transfected into Rat2 cells using calcium phosphate protocol as described previously [47,49] and analyzed 48 h and 72 h post-transfection. The siRNA oligonucleotides targeting FAK#1 (5′-GCTAGTGACGTATGGATGT-3′), FAK#2 (5′-GCUAGUGACGUGUGGAUGUTT-3′); Rack1#1 (5′-AAGGTGTGGAATCTGGCTAAC-3′) and Rack1#2 (5′-GCTAAAGACCAACCACATT-3‘), except FAK#2 siRNA (this study), were described previously [29,40,47]. An additional siRNA was used as a non-specific control (5′-AGGTAGTGTAATCGCCTTG-3′) [40]. siRNA oligonucleotides were synthesized with 3′TT overhangs by Eurofins Genomics (Ebersberg, Germany).

Rat2 cells were transiently transfected either with GFP-VASP (vasodilator-stimulated phosphoprotein), or GFP-Lamin A/C [12] or GFP-LifeAct [40] using Lipofectamine 2000 (ThermoFisher Scientific, Waltham, MA, USA), in accordance with the manufacturer’s protocol. GFP-VASP was amplified from HeLa cells’ cDNA with the primers VASP-F 5′-ttttctcgagtgagcgagacggtcatctgttccagc-3‘ and VASP-R 5′-ttttggatccggtccctgtggtcagggagaaccccg-3‘, and cloned into the GFP-C1 vector using XhoI and BamHI restriction sites.

### 2.3. Replating Assay

Replating experiments were performed essentially as described previously [40]. 48 h and 72 h after siRNA transfection, Rat2 cells (cultivated in DMEM with 10% FBS) were detached by 0.05% trypsin in PBS, treated with trypsin inhibitor (1 μg/mL; Sigma-Aldrich/Merck, Darmstadt, Germany), washed with serum free DMEM and kept in suspension at 37 °C, 5% CO_2_ for 60 min. Cells were then plated on fibronectin (1 μg/mL) coated substrates in a DMEM containing 10% FBS for indicated times. The cells were subsequently used either for live cell imaging or immunofluorescence microscopy.

### 2.4. Western Blot

Cells were washed with ice-cold PBS buffer, lysed in ice-cold RIPA buffer (50 mM Tris–HCl, pH 7.4, 0.1% SDS, 1 mM EDTA pH 8.0, 150 mM NaCl, 1% NP-40, 1% sodium deoxycholate) and clarified by centrifugation at 18,400 g at 4 °C for 20 min. Lysates were boiled in 1 x sample buffer (4% SDS; 20% glycerol; 0.004% bromophenol blue; 0.125M Tris-Cl, pH 6.8; 10% 2-mercaptoethanol) for 5 min, resolved by SDS-PAGE and transferred to nitrocellulose blotting membrane (GE Healthcare, Chicago, IL, USA). Membranes were blocked with 5% non-fat milk in PBS for 1 h at room temperature and then incubated overnight at 4 °C with the primary antibodies. Membranes were probed with HRP-conjugated secondary antibodies and developed using SuperSignal WestPico enhanced chemiluminescent substrate (Pierce, Waltham, MA, USA). The original unprocessed scans of Western blots are shown in Appendix A.

### 2.5. Immunofluorescence and Live Cell Phase Contrast Microscopy

For immunofluorescence staining, cells were cultivated on glass coverslips coated with 1 µg/mL of fibronectin, fixed with 2% paraformaldehyde in 100 mM KPO_4_ (pH 6.5) for 15 min and permeabilized with 0.5–1% Triton X-100 in 100 mM KPO_4_ (pH 6.5) for 10 min. Coverslips were blocked with 1% BSA in 100mM KPO_4_ (pH 6.5), stained with the indicated antibodies, and mounted in Vectashield mounting medium containing DAPI (Vector Laboratories, Burlingame, CA, USA). Fluorescent images were acquired using an Olympus BX43 microscope, 60×/1.35 oil objective, with CMOS Zyla 5.5 (Andor) camera. 

For live cell phase contrast microscopy, cells were cultivated on 6-well plate dishes and images were acquired using an inverted Olympus IX51 microscope, equipped with 10×/0.25 dry objective and CCD F-view II camera (Soft Imaging System).

### 2.6. Coherence-Controlled Holographic Microscopy

For coherence-controlled holographic microscopy (CCHM) of living cells, Rat2 fibroblasts were plated onto glass-bottom petri dishes (Cellvis, Mountain View, CA, USA) and placed in a cultivation chamber (37 °C and 5% CO_2_). Time lapse movies were captured under the same conditions, with the indicated frequencies of the image acquisition. The cells were inspected using a Q-phase microscope (Telight, Brno, Czech Republic), equipped with a pair of 20×/0.5 or 40×/0.95 Nikon Plan Fluor objectives and CCD camera Ximea and CMOS camera Zyla 5.5. Acquired images were processed with inbuilt Q-phase software (Telight) and ImageJ/Fiji (National Institutes of Health). Cell perimeter, area, and dry mass were calculated from segmented quantitative phase images using Q-phase software. Corresponding boxplots were created using RStudio (https://www.rstudio.com). Presented line scans were performed on processed images, i.e., contrast adjusted and blurred with Gaussian filter (Sigma radius = 1), using the Plot profile plugin in ImageJ/Fiji (NIH). 

### 2.7. Quantification of Cell Shape

The cell area and aspect ratio were obtained from phase contrast images of fixed cells in ImageJ by manual marking of the cell periphery. The aspect ratio is the proportion between the major and minor axis of ellipse superimposed on the cell area and indicates cell prolongation. Microscopic images were analyzed and quantified using ImageJ software as reported in [40].

### 2.8. Convexity and Motility Maps

Microscopic images were pre-processed in ImageJ software and analyzed using the QuimP-19.08.01 plugin available from: http://www.warwick.ac.uk/quimp [59,60,61]. QuimP software was used as follows: Cell outlines were extracted from each frame of the image sequence using the BOA plugin (cell segmentation), the movements of cell outlines between individual frames were mapped using the ECMM Mapping plugin, and data analysis was performed by the Q Analysis plugin. The cell edge dynamic was visualized using three spatiotemporal maps. The cell track map is an overlay of cell outlines (with frame increment of 1). The motility map represents the movement of each node on the cell outline over time. The convexity map represents the curvature of the cell outline (in the range −1, 1), where negative values are concave (blue), and positive values convex (red).

## 3. Results

### 3.1. Dry Mass Distribution in Rat2 Fibroblasts Revealed by Coherence-Controlled Holographic Microscopy

We utilized coherence-controlled holographic microscopy (CCHM) and quantitative phase images in order to obtain information about dry mass distribution in cells. Quantitative phase images of Rat2 fibroblasts were reconstructed from acquired hologram images as described previously [58,62]. The signal intensity distribution within the resulting quantitative phase image of a cell corresponds to the dry mass density of the cell’s 2D projection [57] (Figure 1A). Owing to the superior contrast of quantitative phase imaging (QPI) images, cell boundaries can be easily segmented, enabling subsequent measurement of the cell perimeter and area (Figure 1A).

In the dry mass distribution analysis, we biased our analysis to cells that displayed the typical shape of a polarized cell, i.e., an elongated conical shape with a broad front and a tail-like rear. Quantitative phase images of Rat2 cells obtained by CCHM revealed a pattern where the highest cell mass density is at the central part of a cell and gradually decreases toward the cell rear (Figure 1B). The decrease in cell mass toward the cell front was usually sharper and the cell front displayed a low dry mass density. (Figure 1B). The region corresponding to the protruding leading edge (indicated by the yellow arrowhead in Figure 1B) accumulated higher dry mass compared to the transition area behind it. We also observed that high dry mass density sharply decreased at the non-protruding cell sides (Figure 1B, right panels). This profile clearly differs from the dry mass distribution at the leading or trailing edges. 

Since the CCH microscope allows the acquisition of quantitative phase and also fluorescence images, we next correlated the distribution of dry mass with specific cellular compartments labeled by GFP-tagged proteins. We transfected cells with GFP-tagged vasodilator-stimulated phosphoprotein (VASP) that stains protruding membranes and focal adhesions (Figure 1C), GFP-LifeAct to visualize actin (Appendix A), or with GFP-LaminA/C that marks the nuclear envelope (Figure 1D). A line scan analysis of GFP-VASP signal intensity and dry mass density revealed that GFP-VASP localization correlated with the dry mass increase at the leading edge (indicated by the yellow arrowhead in Figure 1C). The GFP-LifeAct signal also correlated with a dry mass increase at the leading edge (Appendix A). In addition, dry mass and GFP-LifeAct localization also correlated at the dorsal side of the leading edge (Appendix A) in structures that resembled peripheral ruffles [63]. It suggests that the protruding edge of migrating cells contains more dry mass then the transition region behind it. Similarly, we correlated cell dry mass distribution with the position of the nucleus labeled by GFP-tagged Lamin A/C. We observed that the nucleus is positioned slightly rearward and that high dry mass density at this region is constituted by the nucleus itself (Figure 1D).

### 3.2. Downregulation of FAK or Rack1 by siRNA Affects Cell Shape and Dry Mass Distribution

We then investigated cell dry mass distribution in cells treated with siRNA targeting the FAK (focal adhesion kinase) or Rack1 (receptor for activated C kinase 1) mRNAs. The siRNA treatment resulted in a significant reduction of both FAK and Rack1 protein levels and the knockdown persisted for at least 72 h (Figure 2A and data not shown). FAK knockdown also diminished FAK protein appearance in focal adhesions (Appendix A). As has been described before [24,40], both FAK and Rack1 protein depletion affected cell shape. FAK depletion resulted in cell elongation and bipolar phenotype characteristics consisting of a central body from which two extensions arise (Figure 2B, “FAK siRNA”). Depletion of Rack1 in Rat2 fibroblasts resulted in a rounded “fried-egg” phenotype (Figure 2B, “Rack1 siRNA”). Quantification of cell shape determined as the aspect ratio confirmed that FAK-depleted cells were elongated while Rack1-depleted cells were rounder than control cells (Figure 2C). 

Using CCHM, we examined dry mass distribution in cells with downregulated expression of FAK or scaffold protein Rack1 (Figure 2D). Line-scan profiles for dry mass distribution revealed significant differences between control and FAK- or Rack1-depleted cells. In the control cells, the highest cell mass density was observed at the central part of the cell and the cell mass gradually decreased toward the cell rear and more abruptly toward the cell front (Figure 2D, “Ctrl siRNA”). In FAK-depleted cells, the highest dry mass density occupied the cell center with the nucleus, from which two distally oriented extensions arise. Cell mass gradually decreased along these extensions toward the cell ends (Figure 2D, “FAK siRNA”), thus resembling the rear end of control cells. The analysis of cell dry mass distribution in Rack1-depleted cells revealed two different phenotypes. The highest dry mass density occupied either the center of the cell (Figure 2D, “Rack1 siRNA”) or the cell periphery where the nucleus was displaced (Appendix A). In both cases, the dry mass sharply decreased from the nuclear region to the basal sides. The dry mass density in the flat area was very low, close to the background values (Figure 2D and Appendix A). Rack1-depleted cells also showed an increase in dry mass at the cell edges in comparison to the flat region, suggesting the existence of protrusions (Figure 2D).

We then quantified the observed differences in cell shape by measuring the cell area and cell perimeter and related these parameters to cell dry mass. Both FAK and Rack1 depletion increased the perimeter of individual cells (Figure 2E and Table 1). The more elongated and circular shapes of FAK and Rack1 cells, respectively, was reflected by the area covered by the individual cells. Cell area was 1.7 times higher in Rack1 cells (median = 2948 µm^2^) compared to FAK cells (median = 1755 µm^2^) (Figure 2E). Quantitative phase imaging allowed us to also measure the dry mass of single Rat2 cells. Control Rat2 fibroblasts had an average (i.e., median) dry mass of 252 pg (Figure 2E). Average dry mass was increased in both FAK- and Rack1-depleted cells by 54% (389 pg) and 131% (583 pg), respectively (Figure 2E and Table 1). 

We also examined whether the simultaneous FAK and Rack1 knockdown reversed the observed phenotypes (Figure 2A,B). In agreement with our previous results [40], we observed that cell elongation was indeed partially rescued in FAK/Rack1 double-depleted cells (Figure 2C). The rescue effect was also indicated by the shape and dry mass distribution in double knockdown cells. These cells were able to form and to spatially segregate broad protruding-like regions and inward oriented cell sides (Figure 2D, “FAK+Rack1 siRNA”). Double depletion of FAK and Rack1 also affected actin organization that resembled more wild type cells, namely, the actin at cell edges (Appendix A). FAK-depleted cells contained thick peripheral stress fibers that run along straight long cell sides. Rack1 suppression displayed radial ovoid symmetry with round cell edges, contained circumferential concentric actin arcs and non-parallel actin bundles, and they were devoid of peripheral actin bundles. In contrast to these two phenotypes, doubly depleted cells displayed inward oriented cell edges underlined by thick peripheral stress fibers (Appendix A). 

On the other hand, the quantification of the cell area, perimeter, and dry mass content revealed that simultaneous depletion of FAK and Rack1 only marginally reversed the phenotype elicited by single depletion of FAK or Rack1. This could be the consequence of cell heterogeneity as the population of cells depleted of both FAK and Rack1 contained cells resembling either FAK- or Rack1-depleted cells. In addition, a large fraction of cells (~ 10%) were represented by huge cells with the dry mass exceeding 1000 pg (Figure 2E). Due to the high heterogeneity, cells depleted of both FAK and Rack1 were not further characterized. 

### 3.3. Spreading Cells with Depleted FAK Develop Two Distally Oriented and Simultaneously Protruding Regions

Interestingly, we occasionally observed that in FAK-depleted cells, the ends of both extensions displayed a significant increase in dry mass density that resembled the protrusions at the leading edge (Appendix A). This observation was surprising as it was previously suggested that in adherent migrating cells, the elongation of FAK-depleted cells is a consequence of impaired retraction of the non-protruding cell rear [24]. We thus examined whether FAK also affects cell shape in cells spreading on fibronectin. Spreading cells were recorded by time-lapse CCHM, followed by comprehensive spatiotemporal mapping of the dynamic and shape of cell edge protrusions, namely cell edge velocity and curvature. To do this, we segmented quantitative phase images using the QuimP toolbox and generated cell outlines for each time point. We determined the cell edge velocity (i.e., protrusions and retractions speed) and curvature (i.e., the shape) from individual points on cell outlines that are linked between contours in time. Cell edge velocity and curvature are presented as 2D spatiotemporal maps showing changes in protrusion speed (or changes in cell edge curvature) along the cell perimeter [59,60,61]. The protruding cell edge is usually convex-oriented as a result of actin pushing forces; thus, there is a correlation between cell protrusions and the convex curvature of the cell edge (see Appendix A and Materials and Methods for details).

Examination of control Rat2 cell spreading on fibronectin revealed that cells initially displayed random protrusions without defined protruding and quiescent edges (Figure 3A,B and Appendix A, time frame 0–10 min). This was followed by the establishment of two distal protruding edges (indicated as regions P1 and P2 in Figure 3A,C, time frame 10–60 min) separated by non-protruding cell regions. Cell edge velocity and curvature analysis showed that the protruding regions adopted positive (convex) curvature while non-protruding regions were either of a straight shape or negatively (concavely) curved (Figure 3C). The protrusions at one pole then gradually ceased while the distal protrusion expanded, ultimately forming a conically shaped cell with a broad protruding front (Figure 3A–C). Interestingly, we also observed that the protrusivity cell poles were to some degree coordinated. The establishment of a protrusion at one pole of the cell was accompanied by reduced protrusions at the opposite pole of the cell (Figure 3C, time frame 30–60 min). Later on, the establishment of a dominant protrusion at the prospective cell front correlated with a gradual decrease in protrusions and convexity at the opposite pole of the cell (Figure 3C, time frame 60–120 min).

The initial phase of polarization of FAK-depleted cells did not grossly differ from control cells as these cells displayed random protrusions (Figure 4A,B, Appendix A, time frame 0–10 min). In the later phase of spreading (time frame 10–60 min), FAK-depleted cells also established two protruding convex regions (indicated as P1 and P2 in Figure 4A,C). However, neither protruding region ceased protruding, rather both cell poles simultaneously protruded. Continuous protrusions then resulted in a bipolar elongated phenotype (Figure 4A–C). It was also apparent that protruding convex regions were narrow compared to control cells (Figure 4C). In addition, FAK-depleted cells frequently developed additional protrusions in the middle of a non-protruding region (indicated by an asterisk in Figure 4A–C).

The simultaneous development of two leading edges represented the prevailing mechanism for establishing the elongated phenotype of FAK-depleted cells. Nevertheless, we also observed that the development of the elongated cell phenotype could be a consequence of a defect in tail retraction. In this case, spreading cells developed a cell front and rear and initiated migration (Figure 4D–F and Appendix A). A migrating cell developed an elongated phenotype as it is unable to retract the trailing edge. At this point, the protrusion was reversed as the cell halted the protrusions at the front and developed a protruding region positioned at the site of the original trailing edge. The protrusions at this site further elongated the cell (Figure 4D–F and Appendix A). 

The spreading control and FAK-depleted cells were further examined by dry mass redistribution analysis. In both cases, the initial phase of spreading was characterized by high dry mass density, reflecting the restricted 2D area covered by the bulk mass of the spreading cell. This phase was followed by a flattening of the cells and establishment of one or two protruding ends in the case of the control and FAK-depleted cells, respectively (Figure 5A,B). Later phase of cell spreading seem to slightly differ between the cell types. The dry mass distribution was slightly shifted to the rear in control cells while it was symmetrical in FAK-depleted cells. In addition, we observed that FAK-depleted cells display dry mass increase at the edges of both elongated processes, supporting the idea that these ends are protruding (Figure 5A,B).

Spreading of Rack1-depleted cells was different from both control and FAK-depleted cells and basically followed the route we described previously [40]. Upon adhesion to fibronectin, Rack1-depleted cells adopted a round shape and omnidirectional protrusions around the cell perimeter. Cells steadily spread with radial symmetry, resulting in an ovoid or discoid shape (Appendix A and Appendix A). 

### 3.4. Focal Adhesions and Actin Organization in FAK-Depleted Cells Spreading on Fibronectin

In polarized mesenchymal cells, the cell sides and rear are characteristics of long, thick stress fibers that underlie the periphery of the cell sides with their ends terminating in large elongated focal adhesions. This organization of focal adhesions and actin gives the cells a typical triangular or conical shape [2]. The inability of FAK-depleted cells to stabilize the cell rear indicated a defect in focal adhesion and/or actin organization at this location. To examine changes in focal adhesions and actin organization in FAK-depleted cells, cells were plated on fibronectin-coated coverslips, fixed at different time points after plating, and stained for actin and focal adhesions’ marker, paxillin (Figure 6). We observed that control cells initially displayed a round shape with a poorly organized actin cytoskeleton and adhesions mainly positioned along the cell periphery (Figure 6A). At later phases of spreading, long stress fibers covered the cell edges and terminated in large focal adhesions at the cell sides. The stress fibers from both the cell’s sides terminated in elongated focal adhesions at the presumed cell rear (Figure 6A, arrowheads) and formed a pointed rear, giving cells the triangular shape. 

Similar to control cells, FAK-depleted cells adopted a bipolar symmetry with the cell sides underlined by stress fibers (Figure 6B). However, there were more stress fibers covering the cell body and they were apparently thicker (Figure 6B), probably due to higher Rho activity in FAK-depleted cells [64]. The focal adhesions at both ends of the cellular extensions were elongated and often parallel (Figure 6B, arrowheads). However, the front and rear cannot be determined as these adhesions do not form a pointed rear. This data indicates that in FAK-depleted cells the formation of a pointed cell rear is compromised, and cells display the 2-fold line symmetry. 

## 4. Discussion

In this work, we used quantitative phase imaging (QPI) based on coherence-controlled holographic microscopy (CCHM) for unbiased characterization of cell morphology and dynamics of FAK- and Rack1-depleted cells. We have primarily focused on the characterization of focal adhesion kinase (FAK) and to lesser extent on Rack1 (receptor for activated C kinase 1), as the depletion of these proteins substantially affects cell shape [24,40]. CCHM is a non-invasive technique for live cell observation generating artifact-free quantitative phase images [65]. Taking into account image reconstruction based on phase interference [66], this approach offers superior visualization even of low-contrast cells. Such QPI images facilitate image processing (i.e., cell segmentation) and subsequent analyses. In addition, dry mass distribution within a cell, or of cells as a whole, can be directly assessed according to the dry mass theory [57]. Using CCHM-QPI, we assessed the dry mass contents of Rat2 cells depleted of either FAK or Rack1. The estimated dry mass contents of Rat2 cells fell well within the range that was determined for HeLa and HT29 cells [57,67]. We observed an increase in the dry mass content of Rat2 cells depleted of either FAK or Rack1. This increase in dry mass did scale up with the cell area. Importantly, we were also able to detect the increase in dry mass at specific intracellular locations, namely at the protruding edge where the increase in dry mass correlated with the localization of GFP-VASP and GFP-LifeAct. Similar to this observation, an increase in dry mass has been observed at the leading edge and pseudopods of cells invading 3D collagen [62]. The increase in dry mass density at the leading edge was associated with membrane protrusion or with structures resembling peripheral ruffles [63]. Hence, ours and other recent reports [62,68,69] stress the potential of label-free non-invasive microscopy in the identification of novel aspects of cell behavior and physiology. 

FAK is a non-receptor tyrosine kinase that serves as a signaling hub in integrin signaling and activates numerous other signaling pathways and cellular processes, namely cell migration. FAK regulates cell migration to a large degree through focal adhesion turnover and trailing edge retraction as these processes are inhibited in FAK-depleted cells [24,43,44,46,70]. FAK also plays a role in the establishment of migratory cell polarity that is a prerequisite for efficient directional cell migration. In wound-healing assays FAK-depleted cells display defects in leading edge organization, microtubule network, and MTOC reorganization and nuclear reorientation [29,49,50,51,52]. This study has revealed a novel function of FAK in the spontaneous establishment of migratory front-rear polarity. By stabilizing the cell rear, FAK promotes the formation of the protruding front and non-protruding cell rear positioned at opposite poles of the cell. 

To examine FAK function, we used the spreading of a cell on extracellular matrix as a simple experimental model of spontaneous cell polarization. During this process the initial round shape of isotropic spreading cells became more irregular as the cells developed a polarized profile with a broad protruding front and a quiescent tail-like rear [38,71,72]. We have previously shown that the appearance of non-protruding concave regions is the first event that breaks the spheroid symmetry of spreading Rat2 fibroblasts. The non-protruding region defines the front-rear polarity axis of polarized cells: The formation of one stable quiescent region gives the cell a crescent-like shape, while two stable concave non-protruding areas define the extended cell rear (Figure 7A) [40]. FAK-depleted cells establish two protruding regions at the cell poles that are separated by non-protruding cell sides. Nevertheless, spreading cells are unable to cease the protrusion at one pole and stabilize the prospective cell rear. Continuous protrusions at the distal regions result in an elongated phenotype with bilateral two-fold rotational symmetry (Figure 7B). This phenotype is strikingly similar to the phenotype of cells spreading on one-dimensional (1D) substrates [42]. Cells spreading on 1D fibronectin lines spread bi-directionally and adopt an elongated phenotype. Cell migration is initiated by the adhesion destabilization at one side of the cell [42]. It is thus plausible to speculate that the elongated phenotype of FAK-depleted cells is the consequence of impaired focal adhesion disassembly. In addition, and in agreement with previous observations of adherent migrating cells [24], we observed that a fraction of spreading cells became elongated as a consequence of rear retraction failure. Thus, the elongated phenotype of FAK-depleted cells could be attained by two different mechanisms, either bidirectional continuous protrusions or impaired cell tail retraction. The dry mass distribution was symmetrical in FAK-depleted cells, in contrast to wild-type cells, suggesting that FAK also affects dry mass distribution in migrating cells. Taken together, these data support the idea that FAK has a pleiotropic function on cell migration, affecting focal adhesion turnover, leading edge organization, trailing edge retraction, the establishment of migratory polarity, and the cell rear specification/determination and stabilization.

How FAK signaling suppresses the formation of multiple leading edges and promotes the establishment of a single leading edge and trailing edge remains unknown. Downstream of integrin engagement to the extracellular matrix (ECM) and integrin activation, FAK is vital in the regulation of actin organization, namely through the temporal regulation of small GTPases of the Rho family [45]. Upon adhesion and integrin engagement to the ECM, Rho activity is temporarily repressed in a FAK-dependent manner and then increases again [64,73]. The decrease of RhoA activity requires p190RhoGAP which associates with FAK, and deletion of both FAK and p190RhoGAP results in elevated RhoA activity [50,74]. The mechanism of increased Rho activity at later points of spreading is not clear but may involve p190RhoGEF and its association with FAK [45]. Our results indicate that Rho activity is indeed upregulated in FAK-depleted cells as these cells contain an increased number of stress fibers and elongated non-protruding cell edges. Notably, Rac activity inversely correlates with RhoA activity: It is high during the initial phase of spreading and then decreases at later time points [75]. Given that Rho and Rac are mutually antagonistic [76,77,78], it is possible that FAK depletion reduces the Rac activity and shifts the equilibrium of Rac/Rho GTPases activities toward Rho. 

Rac is generally thought to regulate Arp2/3 complex-mediated actin polymerization and protrusion formation, and that Rho inhibits this process [7]. These findings are inconsistent with our observation that FAK-depleted spreading cells form protrusions at the cell poles, and with the hypothesis that they display elevated Rho activity. We cannot rule out the possibility that the protrusions at poles of FAK-depleted spreading cells could be driven by residual Rac activity through the Arp2/3 complex confined to these poles. However, we note that cells are able to spread and to form adhesions in the absence of the Rac family of GTPases [79] or in the absence of the active Arp2/3 complex [41,80,81]. In this case, the cells display impaired lamellipodia and utilize alternative filopodial protrusions. In addition, both Rac and Rho are active within the protruding lamellipodium, although the timing and localization of their activation differs during the protrusion cycle [4,5,82,83,84]. Thus, the protrusions can be driven by Rho and its effectors such as formins [82,85], or by the cooperation of formins and the Arp2/3 complex [86]. 

Rac activity and lamellipodial protrusions can be locally suppressed by tension exerted on focal adhesions [87]. Several RhoGEFs such as p190RhoGEF, p115RhoGEF, LARG, and PDZ-RhoGEF localize focal adhesions, and some of them have been shown to associate with FAK [24,88,89]. We thus hypothesize that FAK may also act directly to stabilize the cell rear and reduce the protrusions at this region. It is possible to speculate that by recruiting RhoGEFs to adhesions at the cell rear, FAK may locally increase RhoA activity and actomyosin tension. Tension exerted on focal adhesions would result in adhesion elongation and clustering, and consequently in reduced protrusions without compromising global Rac activity. In agreement with this hypothesis, we observed that during spreading, FAK-depleted cells rarely develop clusters of elongated focal adhesions at the ends. In addition, the protrusions were often seen to develop in the nuclear region (Figure 7B), where the longitudinal stress fibers are prone to breaks due to the presence of a bulky rigid nucleus, furthering the argument that Rac is not globally inhibited in FAK-depleted cells. 

The elongated phenotype of FAK-depleted cells contrasts with the shape of cells depleted of scaffold protein Rack1, as the spreading of the majority of Rack1-depleted cells is dominated by protrusions along the entire cell periphery and isotropic spreading (Figure 7C). Rack1 is a scaffold protein that associates with FAK, and in a FAK-dependent manner regulates activation and localization of several signaling pathways involved in cell polarization and migration [54,56,90]. Rack1 is required for the efficient activation of the ERK pathway which is induced by integrins [47]. We have suggested that Rack1 and ERK promote the development of non-protruding regions by inhibiting p190RhoGAP at the prospective cell rear [40]. Thus, FAK and Rack1 have the capacity, either individually or cooperatively, to coordinate several signaling pathways that control Rho GTPase activity and thus regulate the timing of distinct events during cell spreading. 

## 5. Conclusions

In conclusion, we show that in spreading Rat2 cells, FAK and Rack1 play a significant role in the regulation of the spontaneous establishment of front rear polarity. By utilizing non-invasive and quantitative microscopy, we demonstrate not only dynamic changes in cell shape but also how the dry mass distribution is altered in FAK- or Rack1-depleted cells. Both FAK and Rack1 are required to define the cell rear, and although they associate together, they act at different stages of the polarization. However, it remains to be determined whether the direct interaction of FAK with Rack1 is required for the proper control of cell spreading. It is necessary to establish a more detailed understanding of the crosstalk and co-ordination of the signaling pathways involved in this process.

## Figures and Tables

**Figure 1 biomolecules-10-01089-f001:**
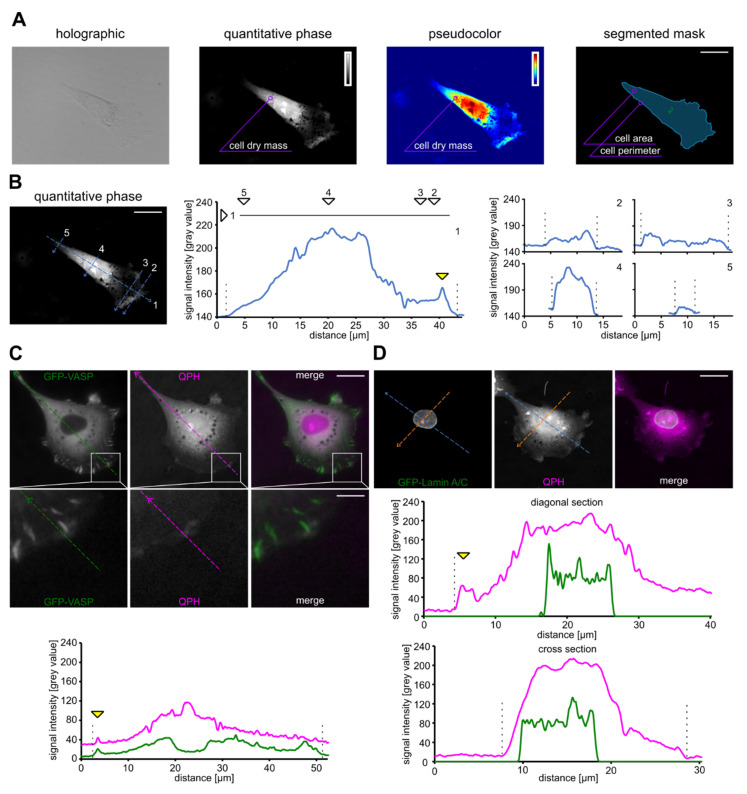
Coherence-controlled holographic microscopy generates quantitative phase images. (**A**) Examples of holographic, quantitative phase, pseudo-colored, and segmented images of a Rat2 fibroblast cell are shown. Cell dry mass distribution is displayed as the grayscale or pseudo-colored quantitative phase images (insets represent the signal intensity scale). Cell area/perimeter can be determined from the segmented mask. (**B**) The graphs show the distribution of signal intensities (i.e., cell dry mass) corresponding to respective diagonal (middle panel) and cross sections (right panel) of the imaged cell. The increase in dry mass at the leading edge is indicated by the yellow arrowhead. Cell borders are indicated by dashed lines. (**C**) GFP-VASP localization and its superposition with quantitative phase images of a Rat2 fibroblast cell are shown. The lower panel displays a higher magnification of the boxed area. The graph shows dry mass distribution and GFP-VASP intensity along the imaged cell. The yellow arrowhead indicates the increase in dry mass and GFP-VASP signal at the leading edge. Cell borders are indicated by dashed lines. (**D**) GFP-LaminA/C localization and its superposition with quantitative phase images of a Rat2 fibroblast cell are shown. The graphs display GFP-Lamin A/C signal intensity and dry mass distribution along the diagonal (blue) section and the cross (orange) section of the imaged cell. The yellow arrowhead indicates the increase in dry mass at the leading edge. Cell borders are indicated by dashed lines. Scale bars 10 µm (**A**,**B**,**C**,**D**), 2.5 µm (C—crop).

**Figure 2 biomolecules-10-01089-f002:**
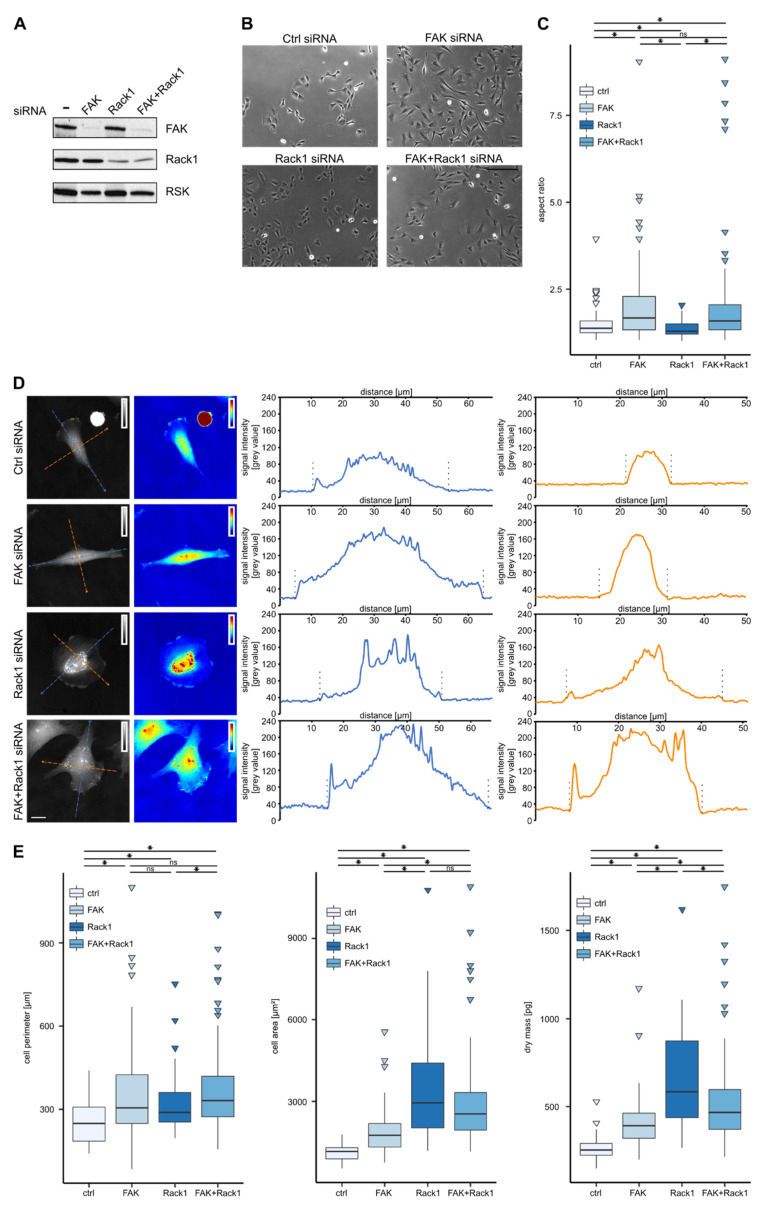
Depletion of FAK and Rack1 triggers two distinct phenotypes depicted by CCHM. (**A**) Expression levels of FAK and Rack1 proteins in control and siRNA treated cells were determined. RSK was used as the loading control. (**B**) Conventional phase contrast microscopy images of the control, FAK, Rack1 and FAK+Rack1 siRNA treated cells are shown. (**C**) Quantification of the cell shape of the control, FAK-, Rack1-, and FAK+Rack1-depleted cells using the aspect ratio (the ratio of the cell length to cell width) is plotted. The representative result of one out of three biological replicates is shown (*n* > 50). (**D**) Grayscale or pseudo-colored quantitative phase images of the control, FAK, Rack1, and FAK+Rack1 siRNA-treated cells are provided (insets represent the signal intensity scale). The graphs show a distribution of signal intensities (i.e., cell dry mass) of corresponding diagonal (blue) and cross (orange) sections. Cell borders are indicated by dashed lines. (**E**) Quantifications of the cell perimeter, cell area, and cell dry mass of the control, FAK, Rack1, and FAK+Rack1 siRNA-treated cells are plotted. The representative result of one out of two biological replicates is shown (n > 50). * *p* < 0.05. Scale bars 200 µm (**B**), 10 µm (**D**).

**Figure 3 biomolecules-10-01089-f003:**
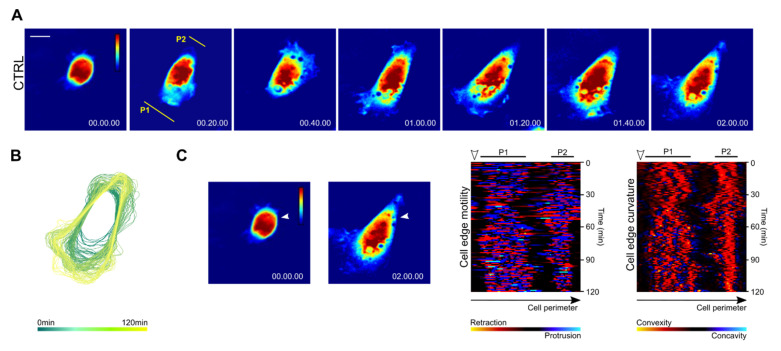
Development of a stable cell rear and front in spreading Rat2 fibroblasts. (**A**) Series of pseudo-colored quantitative images selected from time-lapse CCHM showing protruding and retracting areas in a control Rat2 cell spreading on fibronectin. Time is indicated in h.min.s. P1 and P2 indicate protrusive cell poles 1 and 2, respectively. (**B**) The time sequence of the cell outline during spreading of a control cell superimposed from the first image to the last image. (**C**) Cell edge motility and convexity maps of a spreading cell created in QuimP plugin for ImageJ. The motility map represents changes in the movement speed of the cell edges between two time points. Blue colors indicate expanding regions (protrusions) and red and yellow colors contracting regions (retractions). The convexity map represents the changes in the curvature of the cell edges in time with negative (concave) regions in blue and positive (convex) regions shown in red-yellow. The first and last images of a spreading control cell are displayed in the left panels, white arrows indicate the start point of the cell outline. Scale bar 10 µm.

**Figure 4 biomolecules-10-01089-f004:**
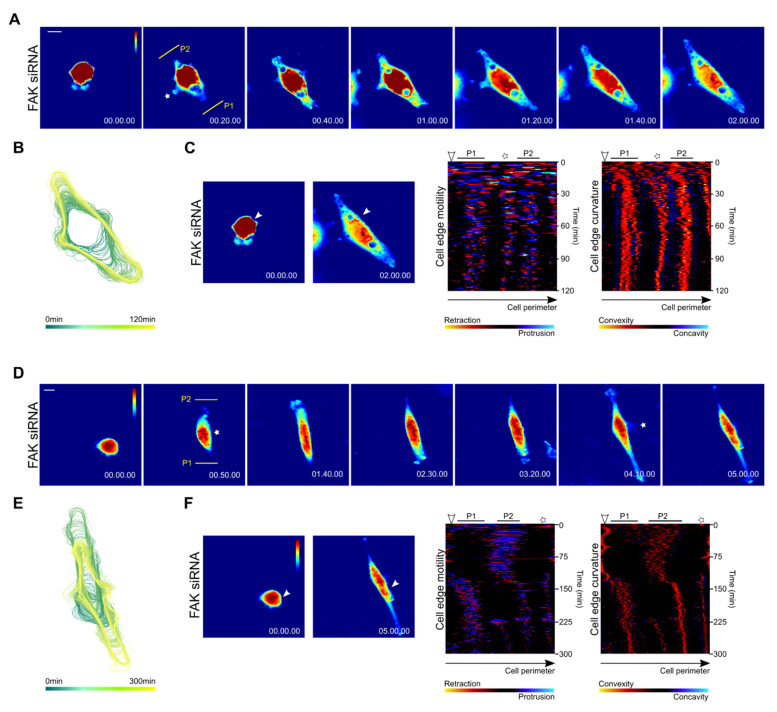
Development of an elongated cell shape in spreading Rat2 fibroblasts depleted of FAK. (**A**,**D**) Series of pseudo-colored quantitative images selected from time-lapse CCHM showing protruding and retracting areas in two FAK-depleted Rat2 cells spreading on fibronectin. Time is indicated in h.min.s. P1 and P2 indicate protrusive cell poles. (**B**,**E**) The time sequences of the cell outline during spreading of two FAK-depleted cells superimposed from the first image to the last image. (**C**,**F**) Cell edge motility and convexity maps of a spreading FAK-depleted cell (see Figure 3C for details). Asterisks indicate a minor protruding region in the nuclear region. Scale bars 10 µm (**A**), 20 µm (**D**).

**Figure 5 biomolecules-10-01089-f005:**
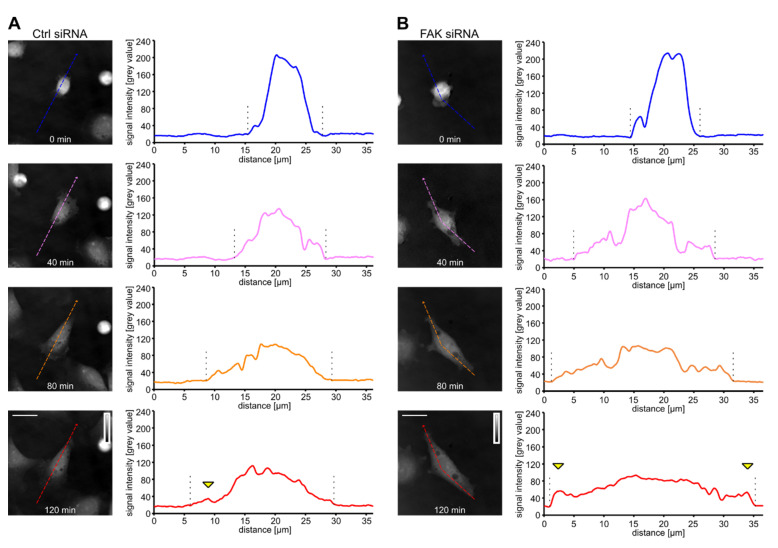
Dry mass distribution changes in spreading control and FAK-depleted cells. Grayscale quantitative phase images of control (**A**) or FAK-depleted (**B**) cells at indicated time points of cell spreading are presented (insets represent the signal intensity scale). The graphs show the distribution of signal intensities (i.e., cell dry mass) of corresponding sections. Cell borders are indicated by dashed lines. The increase in dry mass at a leading edge is indicated by the yellow arrowheads. Scale bar 10 µm.

**Figure 6 biomolecules-10-01089-f006:**
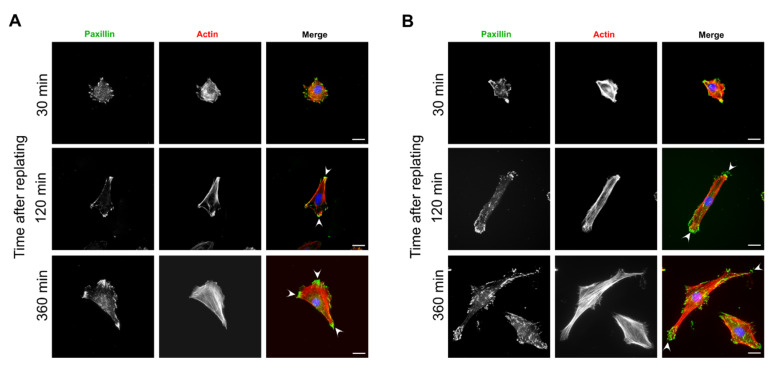
Focal adhesions and actin organization in FAK-depleted cells spreading on fibronectin. Representative fluorescence microscopy images of control (**A**) and FAK deficient cells (**B**) stained for paxillin (green) and actin (red)**.** Rat2 cells were transfected as in Figure 2A and replated on fibronectin-coated coverslips for the indicated times. The arrowhead in (**A**) indicates elongated, clustered focal adhesions at the presumed cell rear of a control cell. The arrowhead in (**B**) indicates focal adhesion organization at the pole of a FAK-depleted cell. Scale bars 20 µm.

**Figure 7 biomolecules-10-01089-f007:**
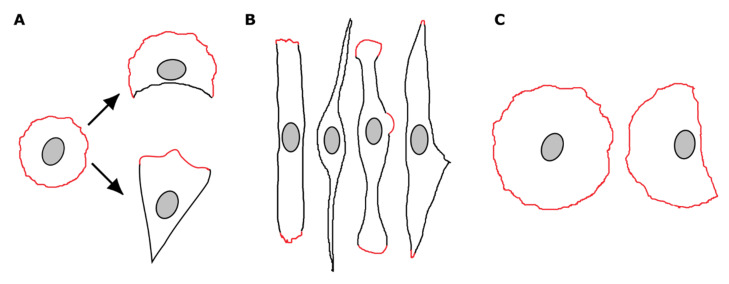
Schematic representation of the shape of Rat2 fibroblasts. (**A**) In radially spreading Rat2 cells, the formation of one non-protruding region results in a crescent-like cell shape while two stable concave non-protruding areas define the extended cell rear. (**B**) FAK-depleted cells adopt an elongated shape with protruding regions at cell poles. Protrusions also often develop at cell sides in the nuclear region. (**C**) Rack1-depleted cells adopt a round phenotype with a centrally located nucleus. A minor fraction of cells adopts a half-round shape with the nucleus displaced to the cell periphery. The protruding regions are highlighted in red, non-protruding cell edges are in black.

**Table 1 biomolecules-10-01089-t001:** Calculated median and corresponding standard error of mean (SEM) values of cell dry mass, area, and perimeter of control and FAK-, Rack1-, or FAK- and Rack1-depleted Rat2 cells.

***Crtl***	**Dry Mass [pg]**	**Area [µm^2^]**	**Perimeter [µm]**
Median	252.3	1145.3	249.9
SEM	8.4	39.1	11.5
***FAK***	**Dry mass [pg]**	**Area [µm^2^]**	**Perimeter [µm]**
Median	389.2	1754.8	305.4
SEM	19.9	104.8	21.7
***RACK1***	**Dry mass [pg]**	**Area [µm^2^]**	**Perimeter [µm]**
Median	582.5	2947.7	290.5
SEM	40.9	308.0	16.6
***FAK + RACK1***	**Dry mass [pg]**	**Area [µm^2^]**	**Perimeter [µm]**
Median	466.8	2535.8	332.3
SEM	34.3	251.2	22.6

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
