# Peer review of "Quantitative Phase Imaging of Spreading Fibroblasts Identifies the Role of Focal Adhesion Kinase in the Stabilization of the Cell Rear"

_biomolecules, 2020, doi:10.3390/biom10081089_

Round 1
Reviewer 1 Report
In this report, Ramaniuk et al. carried out experiments to demonstrate that FAK and RACK1 are involved in formation of a sable rear using knockdown of the proteins using siRNA. Overall, the experiments are relatively well done with appropriate controls and statistical analyses. Especially, the morphological changes of cells are apparent after knockdown of the proteins, FAK and Rack1. However, in addition to the specific comments listed below, the major issue I have with this work is that it is highly limited in scope, in my opinion, and this work does not address in any depth the potential biological significance of these findings. In addition, panels in the figures are not concise but very redundant.
Other comments.
- It is better to perform double knockdown of the proteins to see the relationship in formation of a stable rear
- Overexpression of the proteins will be an alternative approach to evaluate the effects of the proteins on formation of a stable rear.
- Rat2 cells are derived from Rats, thus RACK1 should be Rack1.
- ‘Dry mass’ is very strange to me. It is good to introduce what ‘dry mass’ is and its meaning and significance in the introduction part.
Author Response
Response to Reviewers:
We would like to thank the reviewers for their comments and helpful analysis of our manuscript which we believe has strengthened our paper. In the revised manuscript we incorporated new results
(Figure 2, Figure S1, Figure S4, Figure S6 and Table1) and rewrote the text accordingly. We also replaced Figure 1D with a more representative image. Major corrections are highlighted in the manuscript.
Both reviewers raised the question about the biological significance of our findings. We hence rewrote and extended some parts of the manuscript, the Introduction and Discussion sections in particular, to make the text more clear and to emphasize our findings in the context of cell polarization and cell migration.
Reviewer 1.
Comment: The major issue I have with this work is that it is highly limited in scope, in my opinion, and this work does not address in any depth the potential biological significance of these findings.
Response: Spontaneous establishment of cell asymmetry (also referred to as spontaneous symmetry breaking) is involved in fundamental processes, such as cell motility, embryonic development and morphogenesis. However, the signaling pathways that control spontaneous symmetry breaking and polarization of adhering cells still remain incompletely understood. We believe that the identification of the central players of this process such as FAK and Rack1, and elucidation of their effect on cell polarization, is relevant especially in the context of cell migration. We expanded the Introduction and Discussion sections to emphasize the biological significance of our results.
Comment: In addition, panels in the figures are not concise but very redundant.
Response: We believe that the reviewer is referring to Figure 3, Figure 4 and Figure S7. These figures indeed contain some panels that can be considered redundant. However, we think that these panels can guide the reader through quite complex analyses of the cell edge velocity and curvature. To support this opinion, Reviewer 2 asked for a more detailed description of cell edge analyses (that are now included in supplementary Figure S6).
Other comments:
Comment 1: It is better to perform double knockdown of the proteins to see the relationship in formation of a stable rear.
Answer: As suggested by this reviewer, we performed simultaneous knockdown of FAK and Rack1. In agreement with our previous report (Klimova et al., 2016) we found that the changes in cell shape (i.e. cell elongation) of FAK or Rack1 single depleted cells were partially reverted by double FAK/Rack1 knockdown. On the other hand, double FAK/Rack1 has not reverted all the morphological changes as it displayed only a small or no effect on the average cell size or cell dry mass (when compared to single knockdown cells). We also noted cell heterogeneity in the double knockdown cell population as it contained cells resembling wild-type, FAK or RACK1 depleted cells, and also a significant fraction of extremely large cells. This data is now provided in the revised manuscript (Figure 2, Figure S4 and Table 1).
Since the double knockdown of FAK and Rack1 has so deleterious an effect on cell morphology and the cell population was very heterogeneous, these cells were not further characterized.
Comment 2: Overexpression of the proteins will be an alternative approach to evaluate the effects of the proteins on formation of a stable rear.
Response: The high level over-expression of the proteins of interest, namely Rack1, proved to be beyond reach technically. For unknown reason, Rat2 cells express only miniscule amounts of Rack1 without significant changes in morphology. Lentiviral mediated gene delivery was also unsuccessful as it seems that the infection itself changes the morphology of Rat2 cells. As for FAK, it has been reported before (Iwanicki et al., 2008) that FAK overexpression in Rat2 cells does not visibly affect cell shape and elongation; although at the same time FAK overexpression can rescue the cell elongation defect in cells where endogenous FAK was depleted.
Comment 3: Rat2 cells are derived from Rats, thus RACK1 should be Rack1.
Response: Thank you for noticing the error. The protein name has been corrected.
Comment 4: ‘Dry mass’ is very strange to me. It is good to introduce what ‘dry mass’ is and its meaning and significance in the introduction part.
Response: The concept of dry mass is now explained in more detail in the Introduction section of the revised manuscript. Specifically, we highlight the fact that the quantitative phase imaging technique enables the monitoring of the phase distribution (i.e. phase shift) of a light wave upon passing through a cell. Since the phase shift is not affected by the cellular water content (Popescu et al., 2008), the acquired signal is directly proportional to the cell dry mass, i.e. the non-aqueous content of a cell (i.e. proteins, nucleic acids, lipids, sugars etc.).
Reviewer 2 Report
The authors in the manuscript try to follow and visualise role of FAK and ROCK1 in cell motility in case of Rat2 cells. The text is well written, language is adequate, grammar mistakes can be hardly found. I do support the publication of the manuscript after minor explanations and corrections are done which are listed below.
Questions, comments and suggestions:
Why are all images ranging in pixels instead of more intuitive distance measures of microns? Since there is a calibration for all images, it is easy to convert the scales. THinking in microns is more convenient and intuitive, than in pixels.
What is exaclty the difference between protrusion maps and Convexity maps? They seem strongly correlated. Some detailed explanation about their calculation wouold be benefical to the understanding for non field related reader (even in the supplement in case there are size limitations of manuscript)
Fig 5B. While dry mass profiles in case of control are drawn in a straight line, for FAK depleted cells they are drawn in an agle? What is the reason for that? Would it change the results of the profile is drawn in a line?
Supplementary vidoes: It is not clear what are the green circles on the right panel (cell contours reconstruction) of the video sequence. How were the dashed borders defined and calculated? Some more explanation would be benefical for readers not familiar with these representations.
Role of FAK and ROCK1 are well described in the literature in many cell types. The authors should emphasise and highlight the contribution and novelty of the manuscript.
Author Response
Response to Reviewers:
We would like to thank the reviewers for their comments and helpful analysis of our manuscript which we believe has strengthened our paper. In the revised manuscript we incorporated new results
(Figure 2, Figure S1, Figure S4, Figure S6 and Table1) and rewrote the text accordingly. We also replaced Figure 1D with a more representative image. Major corrections are highlighted in the manuscript.
Both reviewers raised the question about the biological significance of our findings. We hence rewrote and extended some parts of the manuscript, the Introduction and Discussion sections in particular, to make the text more clear and to emphasize our findings in the context of cell polarization and cell migration.
Reviewer 2.
Comment: Why are all images ranging in pixels instead of more intuitive distance measures of microns? Since there is a calibration for all images, it is easy to convert the scales. Thinking in microns is more convenient and intuitive, than in pixels.
Response: Thank you for pointing this out. The dimensions in the graphs were converted from pixels to microns.
Comment: What is exactly the difference between protrusion maps and Convexity maps? They seem strongly correlated. Some detailed explanation about their calculation would be beneficial to the understanding for non-field related reader (even in the supplement in case there are size limitations of manuscript).
Response: The reviewer is correct that cell edge velocity (protrusions) and cell edge positive curvature (convexity) are correlated. This is due to actin polymerization: forces mediated by actin polymerization push on the cell membrane to induce its forward movement (i.e. cell protrusion). At the same time the protruding regions adopt positively shaped edges. The explanation is included in the revised manuscript. In the supplementary material (Figure S6) we now also provide a more detailed description of how cell edge velocity (protrusions and retractions) and cell edge shape (convexity) are determined.
Comment: Fig 5B. While dry mass profiles in case of control are drawn in a straight line, for FAK depleted cells they are drawn in an angle? What is the reason for that? Would it change the results of the profile is drawn in a line?
Response: During spreading of FAK depleted cells, the cell body quite often becomes bent because of the break in one actin cable underlining the cell side. As we intended to show the dry mass distribution along the longest path of the cell (i.e. connecting the ends of elongated processes) and since it is not possible to draw the same straight line across the entire cell at all the time points indicated, we connected the ends by a broken line.
Interestingly, the sites of actin fiber breakage are also the sites of actin polymerization and protrusion formation, indicating that the integrity of thick peripheral cables is antagonistic to cell protrusivity.
Comment: Supplementary videos: It is not clear what are the green circles on the right panel (cell contours reconstruction) of the video sequence. How were the dashed borders defined and calculated? Some more explanation would be beneficial for readers not familiar with these representations.
Response: The green circles indicate the sites where the cell outlines from two different time points intersect. Thus, green circles demarcate the regions that are either protruding or retracting.
The methodology of cell edge velocity (protrusion and retraction speed) and convexity (cell edge shape) mapping was described previously in references [59-61]. As noted above, in the supplementary material (Figure S6 and accompanying figure legend) we provide a more detailed description of how cell edge velocity and shape is determined.
Comment: The authors should emphasize and highlight the contribution and novelty of the manuscript.
Response: As described in response to Reviewer 1, the topic of this manuscript deals with spontaneous establishment of cell asymmetry (also referred to as spontaneous symmetry breaking) that is involved in fundamental processes, such as cell motility, embryonic development and morphogenesis. However, the signaling pathways that control spontaneous symmetry breaking and polarization of adhering cells still remain incompletely understood. We believe that the identification of the central players of this process such as FAK and Rack1, and elucidation of their effect on cell polarization, is relevant especially in the context of cell migration. We expanded the Introduction and Discussion sections to emphasize the biological significance of our results.
Round 2
Reviewer 1 Report
The authors sincerely addressed all comments raised. I believe that the work, although limited in scope, advances the field in a meaningful way. I recommend publication of the manuscript in the journal.